# FusionFormer: Multi-Window Fusion for Efficient Real-Time Segmentation with Vision Foundation Models

## Abstract

Recent advances in real-time segmentation has been driven by lightweight Transformer variants. To address the limitations of such designs, state-of-the-art (SOTA) methods employ bidirectional architectures with training-only Transformer branch for long-range contextual guidance. However, these approaches typically depend on task-specific pre-trained models with limited scalability, potentially limiting their maximum performance. In this paper, we introduce the *Multi-Window Fusion Transformer* (**FusionFormer**) to effectively leverage vision foundation models (VFMs). Specifically, we constrain self-attention computation within different window sizes and aggregate tokens within each window using varying fusion ratios. Our attention mechanism approximates attended fields from local to global while maintaining linear computational complexity, enabling better utilization of global attention guidance in Vision Transformer (ViT)-based VFMs. We first evaluate FusionFormer on the ImageNet-1k classification task, demonstrating its potential as a versatile efficient backbone. Further, extensive experiments on the ADE20K and Cityscapes datasets, coupling FusionFormer with a simple light head and DINOv2-B/14, demonstrate its excellent trade-off between segmentation accuracy and computational cost. Compared to previous methods, FusionFormer achieves a more efficient utilization of global guidance from VFMs. Our code will be released soon after the paper is accepted.

## 1 Introduction

As a representative dense prediction task, semantic segmentation aims to assign a semantic category to each pixel, demanding fine-grained perception while maintaining global consistency. It significantly contributes to advancements in autonomous driving Muhammad et al. (2022), medical image processing Qureshi et al. (2023), remote sensing Lv et al. (2023). With the success of ViT Dosovitskiy et al. (2020) in various vision tasks, ViT-based segmentation models Zheng et al. (2021); Strudel et al. (2021); Zhang et al. (2022a; 2024a) have demonstrated promising performance in general semantic segmentation.

Despite the rich semantic representations through self-attention mechanisms, the quadratic computational cost hinders their application in real-time segmentation. To address these limitations, recent works have explored lightweight Transformer variants that restrict the scope of self-attention for efficiency, either through window-based attention Huang et al. (2019a); Vaswani et al. (2021); Wang et al. (2021); Liu et al. (2021) as in HRFormer Yuan et al. (2021), or by limiting attention to high-stride stages as in TopFormer Zhang et al. (2022b) and RTFormer Wang et al. (2022a). SeaFormer Wan et al. (2023) builds on Axial attention Huang et al. (2019b); Ho et al. (2019); Wang et al. (2020a) to achieve an excellent trade-off between performance and efficiency. However, its aggressive compression of attention computation limits global context modeling and results in suboptimal accuracy. SCTNet Xu et al. (2024) integrates convolutional attention Guo et al. (2022); Chen et al. (2020) with a training-only Transformer branch for semantic guidance, thereby achieving high efficiency and improved accuracy without increasing inference cost.

However, we observe that SCTNet and subsequent works Yin et al. (2024); Dong et al. (2025); Wan et al. (2025) primarily rely on task-specific pre-trained segmentation models such as SegFormer-B2

and SegFormer-B3 Xie et al. (2021), which enable guidance during both encoding and decoding stages. This reliance constrains overall performance and generalization, a limitation highlighted when considering powerful VFMs such as DINOv2 Oquab et al. (2023). Even when using frozen features and a simple linear decoder, DINOv2 surpasses a wide range of real-time segmentation methods and commonly used guidance models, especially on ADE20K with 150 fine-grained semantic categories. For example, DINOv2-S/14, with 21M parameters, achieves **47.2%** mIoU, outperforming SeaFormer-L (43.7%), SCTNet-B (43.0%), and SegFormer-B2 (46.5%). Its larger counterpart, DINOv2-B/14, reaches **51.3%**, exceeding SegFormer-B5 (51.0%) with comparable size. Although DINOv2 underperforms in high-resolution tasks like Cityscapes due to the coarse linear decoders and aggressive upsampling, its remarkable semantic representations raise an important question: ***Can we fully leverage vision foundation models to guide real-time segmentation models and achieve superior performance?***

In this paper, we propose a new lightweight Transformer variant, the *Multi-Window Fusion Transformer* (**FusionFormer**), which approximates global attention by capturing dependencies across short to long ranges, aligning more effectively with the guidance from ViT-based foundation models. The core component of FusionFormer, Multi-Window Fusion Attention (MWFA), captures multi-scale interactions via sliding windows of varying sizes, while assigning local fusion ratios within each window to reduce redundancy and computational overhead. For local token fusion with a ratio of $r$, we concatenate queries, keys, and values and apply a depth-wise convolution to generate adaptive fusion weights. Each $r^2$ region within different attention heads is individually enhanced, allowing fine-grained modulation that compensates for detail sacrificed during fusion. Regarding the guidance process, our method leverages VFMs by jointly utilizing the fine-grained patch tokens, which capture localized semantic details, and the global class token, which provides holistic scene-level context, thereby enabling both spatial precision and semantic richness in guiding segmentation.

In a nutshell, our contributions can be summarized as follows: **1)** We propose *FusionFormer*, a lightweight Transformer tailored for real-time semantic segmentation. **2)** We design *Multi-Window Fusion Attention* to approximate global context with high efficiency via multi-scale sliding windows and adaptive token fusion. **3)** We demonstrate FusionFormer's strong performance on ImageNet-1K classification, validating its versatility as a general-purpose vision backbone. **4)** We develop a guidance mechanism that effectively injects semantic priors from VFMs by combining patch and class tokens. **5)** Our method achieves SOTA performance on ADE20K and Cityscapes with an excellent accuracy–efficiency balance.

## 2 METHOD

### 2.1 PRELIMINARY

**Motivation.** We aim to enhance the performance of lightweight segmentation models through the guidance of ViT-based foundation models, which offer superior fine-grained semantic perception and global contextual guidance. However, mainstream lightweight designs often overly sacrifice global interaction patterns, fundamentally diverging from the global attention paradigm of ViTs. Our goal is to approximate the global receptive field while maintaining computational efficiency, thereby facilitating acquisition of rich semantic representations from VFMs.

**Vision Foundation Models.** Through large-scale pretraining on massive datasets, VFMs Radford et al. (2021); Caron et al. (2021); He et al. (2022); Zhou et al. (2021); Oquab et al. (2023) provide general and semantically rich visual features that benefit various downstream tasks. In particular, recent advances in self-supervised learning (SSL), exemplified by masked image modeling Bao et al. (2021); Wei et al. (2022); He et al. (2022); Zhou et al. (2021), have yielded impressive results when fine-tuned for dense prediction tasks. Notably, DINOv2 Oquab et al. (2023) builds upon the student–teacher architecture Caron et al. (2021), where the teacher network is updated via an exponential moving average (EMA) of the student. Its training combines image-level contrastive learning (via multi-crop class token alignment) with patch-level masked image modeling for global crops, inspired by iBOT Zhou et al. (2021). Pretrained with this comprehensive SSL design on the curated and diverse LVD-142M dataset, DINOv2-g/14 achieves SOTA performance across a wide range of benchmarks, which is then utilized as a frozen teacher within a unified training framework to guide the development of smaller variants. Thereby, various DINOv2 variants demonstrate strong generalization performance across diverse dense prediction tasks with frozen features.

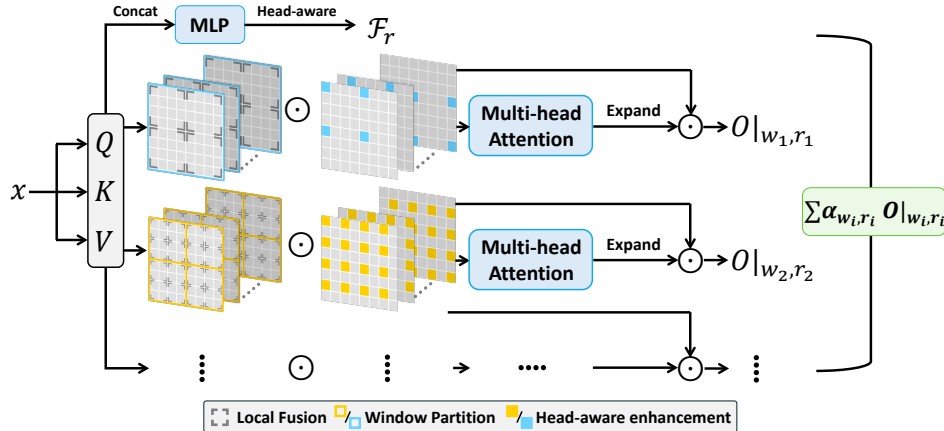

Figure 1: The architecture of MWFA, including the adaptive token fusion with head-aware enhancement and the multi-window fusion, where $\odot$ denotes the Hadamard product.

## 2.2 MULTI-WINDOW FUSION ATTENTION

Transformers establish long-range dependencies through self-attention, which is typically formulated as follows:

$$O = \text{Softmax}\left(\frac{QK^{\top}}{\sqrt{d_{qk}}}\right)V \tag{1}$$

where $Q, K \in \mathbb{R}^{N \times d_{qk}}$ and $V \in \mathbb{R}^{N \times d_v}$ denote the query, key, and value tensors and $d_{qk}$, $d_v$ denote their corresponding attention head's dimension. Global self-attention incurs quadratic complexity in the number of patch tokens $N = HW$ ($\mathcal{O}(N^2)$), leading to inefficiency in dense prediction tasks.

Window-based attention Huang et al. (2019a); Vaswani et al. (2021) restricts self-attention computation to non-overlapping local windows, reducing its cost to $\mathcal{O}(Nw^2)$ with window size $w$. Even with improved mechanisms like shifted windows Liu et al. (2021) and varying windows Yan et al. (2024); Zhang et al. (2024b), it still suffers from the inherent limitation of a fixed and small window size (e.g. $w = 8$). However, simply enlarging $w$ leads to quadratic growth in computational cost and risks losing the focus on local details.

To approximate the global attention field while maintaining efficiency, we propose an attention mechanism called Multi-Window Fusion Attention (MWFA). As shown in Figure 1, MWFA is designed to capture dependencies ranging from short to long-range interactions by combining multiple windows, enabling both fine-grained local detail modeling and long-range contextual awareness.

**Window Fusion Attention.** For a specific window size $w$, the input ($Q$, $K$, and $V$) are split into non-overlapping windows $\{\tilde{Q}_i, \tilde{K}_i, \tilde{V}_i\}^{\frac{HW}{w^2}}$. As the computational cost increases with $w^2$, we observe that token interactions within larger windows often introduce redundancy, since fine-grained local details have already been captured by smaller-window attention.

According to this drawback, we introduce a local fusion mechanism that adaptively aggregates tokens within each window. As illustrated in the left path of Figure 1, the projected $Q$, $K$, and $V$ are concatenated to generate fusion weights for all windows, expressed in Equation 2.

$$\mathcal{F} = \text{Sigmoid}\left(\psi\left(\text{Concat}(Q, K, V)\right)\right) \tag{2}$$

where $\psi(\cdot)$ denotes a convolution-based MLP, and $\mathcal{F} \in \mathbb{R}^{n_h \times H \times W}$ corresponds to the separate fusion weights for each attention head.

Given the fusion ratio $r$ for current $w$, we further enhance $\mathcal{F}$ by exploiting the independence of attention heads. Specifically, to encourage spatial diversity across attention heads, we introduce a head-aware enhancement mechanism. For the $h$-th attention head, we apply a fixed bias to the $h$-th token within the $r^2$ flattened tokens of each window, as defined in Equation 3:

$$\mathcal{F}_{r,j}^{(h)} = \mathcal{F}_{r,j}^{(h)} + \delta(j = h) \tag{3}$$

where $\delta(\cdot)$ is an indicator function, and $j \in \{1, \ldots, r^2\}$ denotes the index of the token within the fusion region $\mathcal{N}^{r^2}$. Accordingly, the computation under the combination of $(w, r)$ is defined as follows:

$$\tilde{Q} = \mathcal{F}_r \odot Q, \tag{4}$$

$$\tilde{Q}_i = \left[ \sum_{j \in \mathcal{N}_{i,0}^{r^2}} \frac{\tilde{Q}_{i,j}}{r^2}, \sum_{j \in \mathcal{N}_{i,1}^{r^2}} \frac{\tilde{Q}_{i,j}}{r^2}, \cdots, \sum_{j \in \mathcal{N}_{i,\frac{w^2}{r^2}}^{r^2}} \frac{\tilde{Q}_{i,j}}{r^2} \right] \tag{5}$$

$$\tilde{O}_i = \text{Softmax}(\frac{\tilde{Q}_i \tilde{K}_i^T}{\sqrt{d_{qk}}} \tilde{V}_i) \tag{6}$$

$$O_i = \mathcal{F}_r \odot \text{Expand}(\tilde{O}_i) \tag{7}$$

where $\mathcal{N}_{i,j}^{r^2}$ denotes the $j$-th fusion region of $\tilde{Q}_i$, $\odot$ denotes Hadamard product and $\tilde{K}_i, \tilde{V}_i$ all follow the same process in Equation 4 and Equation 5. The attention output $\tilde{O}_i$ (from Eq. equation 6) is then expanded back to the original window resolution. As defined in Equation 7, this expansion is guided by the fusion weights $\mathcal{F}_r$ to preserve the spatial distribution imposed during fusion.

**Multi-Window Fusion.** In practice, the window size $w$ trades the globality of attention for efficiency, while the fusion ratio $r$ further reduces the computation complexity to $\mathcal{O}(N\frac{w^2}{r^4})$. To approximate global attention more efficiently across spatial scales, MWFA employs a mixture of different window sizes and fusion ratios $\{(w_i, r_i)\}_{i=1}^k$, leading the complexity to $\mathcal{O}(N \sum_{i=1}^k \frac{w_i^2}{r_i^4})$. The final output is a weighted sum of the outputs from each $(w_i, r_i)$ configuration:

$$O = \sum_{i=1}^k \alpha_i \cdot O_{w_i, r_i}$$

where the weight $\alpha_i$ is computed as:

$$\alpha_i = \frac{s_i}{\sum_{j=1}^k s_j}, \quad \text{with} \quad s_i = \text{LSE}(A_{w_i, r_i})$$

Here, $A_{w_i, r_i}$ denotes the attention scores corresponding to the configuration $(w_i, r_i)$, and $\text{LSE}(\cdot)$ refers to the log-sum-exp operation. This weighting encourages the model to emphasize the most confident attention patterns from different spatial contexts.

**Unified position embedding.** Relative position bias (RPB), widely adopted in window-based attention mechanisms Liu et al. (2021; 2022), effectively models the spatial relationships among tokens within local windows. However, such embeddings are inherently limited to intra-window interactions and lack global spatial awareness for multi-window settings. Inspired by Rotary position embedding (RoPE) for 2D images Su et al. (2024b); Heo et al. (2024), we apply a unified 2D RoPE embedding to the specific stage, enabling spatial relationships across different window partitions.

## 2.3 OVERALL ARCHITECTURE

To leverage the guidance of VFMs while maintaining computational efficiency, we propose a *Multi-Window Fusion Transformer* (**FusionFormer**) framework. As shown in Figure 2, we introduce our method from these parts: *FusionFormer backbone*, *light head*, and *guidance head*.

**FusionFormer Backbone.** To ensure fair comparison with prior lightweight segmentation methods Zhang et al. (2022b); Wang et al. (2022a); Xie et al. (2021); Xu et al. (2024), FusionFormer's STEM features three stages constructed with DSConv Howard (2017) and MBConv Sandler et al. (2018) blocks, yielding feature maps at strides of 4, 8, and 16. Stage 1 achieves rapid early downsampling to a stride of 4 using an initial standard convolution (stride 2) and a DSConv, followed by two MBConv blocks (the first also with stride 2). Stages 2 and 3 then further downsample the features to strides of 8 and 16 respectively, each employing two MBConv blocks where the first provides stride-2 downsampling. Beyond the STEM, the stride 16 features feed into two subsequent stages composed of FusionFormer layers. These layers utilize MWFA to capture dependencies from

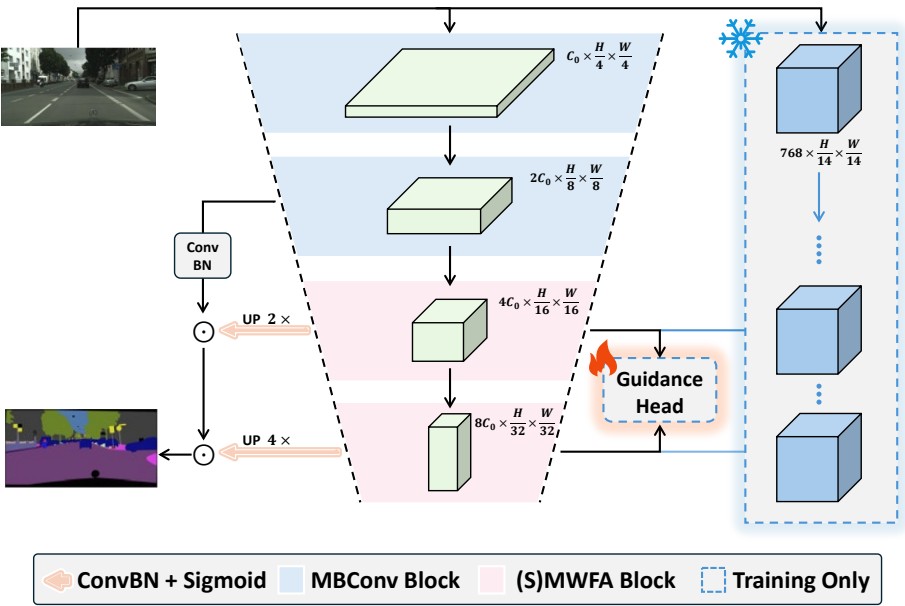

Figure 2: The overall architecture of FusionFormer. The middle path shows the FusionFormer backbone with STEM and (S)MWFA blocks. The right path illustrates the frozen DINOv2-B/14 and trainable guidance head, while the left path shows the light head.

local details to long-range contextual cues, thereby constructing rich semantic representations. The internal computation of successive (S)MWFA blocks can be summarized as:

$$\hat{x}^{l+1} = \text{(S)MWFA}(\text{LayerNorm}(x^l)) + x^l,$$
$$x^{l+1} = \text{MLP}(\text{LayerNorm}(\hat{x}^{l+1})) + \hat{x}^{l+1}$$

where SMWFA denotes MWFA with shifted window Liu et al. (2021), and MLP refers to a two-layer feed-forward network with GELU activation. Note that our MLP utilizes standard linear layers, differing from prior works Zhang et al. (2022b); Guo et al. (2022); Wan et al. (2023); Xu et al. (2024); Cao et al. (2024) that often use $3 \times 3$ convolutions.

We unify the number of FusionFormer blocks in the last two stages to $\{6, 2\}$, and apply an additional MBConv block in the final stage to further downsample the feature to a stride of 32. By varying the embedding dimensions, we derive three model variants: FusionFormer-Small (S), FusionFormer-Base (B), and FusionFormer-Large (L). The detailed configurations are provided in the Appendix A.1.

**Light Head.** To fully exploit the multi-scale features without additional decoding overhead, we adopt a lightweight fusion strategy based on Hadamard product. Instead of aggregating a wide scale of features (e.g., stride from 8 to 64) Zhang et al. (2022b); Wan et al. (2023; 2025), we selectively inject only the last two stages into the stride-8 feature map. Compared to SCTNet Xu et al. (2024) that adopts a DAPPM head, and SDPT Cao et al. (2024) that enhances decoding with attention, we achieve a more efficient yet effective decoding.

**Guidance Head.** Since MWFA is designed to approximate global attention from local to long-range interactions, the features from the last two stages can be naturally aligned with the guidance from ViT-based VFMs. We discard the use of hard supervision from decoder logits Xu et al. (2024); Wan et al. (2025); Huang et al. (2019b) and instead focus on leveraging frozen features from VFMs, including fine-grained patch tokens and the globally aggregated class token. For fine-grained guidance from patch tokens, features from FusionFormer are first projected to match the embedding dimension of the ViT using DSConv, and then upsampled to align with the spatially uniform patch tokens. We apply the channel-wise distillation (CWD) loss Shu et al. (2021) to learn spatial distributions across feature dimensions. For global semantic guidance, we first align the channel-wise distributions of the patch tokens with the reshaped features from FusionFormer. The alignment process is

Table 1: ImageNet-1K validation results. 'Acc.' indicates Top-1 accuracy. FLOPs are measured with 224×224 input size.

| Method | Reference | Params (M) | FLOPs (G) | Acc (%) |
|---|---|---|---|---|
| MiT-B0 Xie et al. (2021) | NeurIPS 2021 | 3.7 | 0.6 | 70.5 |
| Mobile-Former96 Chen et al. (2022) | CVPR 2022 | 4.8 | 0.1 | 72.8 |
| PVTv2-B0 Wang et al. (2022b) | CVM 2022 | 3.4 | 0.6 | 70.5 |
| MobileOne-S0 Vasu et al. (2023) | CVPR 2023 | 2.1 | 0.3 | 71.4 |
| SeaFormer-S Wan et al. (2023) | ICLR 2023 | 4.1 | 0.2 | 73.3 |
| SDPT-Tiny Cao et al. (2024) | TITS 2024 | 3.2 | 0.6 | 71.9 |
| FusionFormer-S | Ours | 3.8 | 0.6 | **73.9** |
| Mobile-Former151 Chen et al. (2022) | CVPR 2022 | 7.7 | 0.2 | 75.2 |
| MobileViT-S Mehta & Rastegari (2021) | ICLR 2022 | 5.6 | 0.5 | 75.2 |
| EdgeViT-XS Pan et al. (2022b) | ECCV 2022 | 6.7 | 1.1 | 77.5 |
| MobileOne-S2 Vasu et al. (2023) | CVPR 2023 | 7.8 | 1.3 | 77.4 |
| SeaFormer-B Wan et al. (2023) | ICLR 2023 | 8.7 | 0.3 | 76.0 |
| FusionFormer-B | Ours | 7.9 | 1.3 | **79.3** |
| MiT-B1 Xie et al. (2021) | NeurIPS 2021 | 14.0 | 2.1 | 78.7 |
| Mobile-Former508 Chen et al. (2022) | CVPR 2022 | 14.8 | 0.5 | 79.3 |
| PVTv2-B1 Wang et al. (2022b) | CVM 2022 | 13.1 | 2.2 | 78.7 |
| P2T-Tiny Wu et al. (2022) | TPAMI 2022 | 11.6 | 1.8 | 79.8 |
| EdgeViT-S Pan et al. (2022b) | ECCV 2022 | 11.1 | 1.9 | 81.0 |
| MobileOne-S4 Vasu et al. (2023) | CVPR 2023 | 14.8 | 3.0 | 79.4 |
| SeaFormer-L Wan et al. (2023) | ICLR 2023 | 14.0 | 1.2 | 79.9 |
| SDPT-Small Cao et al. (2024) | TITS 2024 | 11.6 | 1.9 | 80.6 |
| FusionFormer-L | Ours | 13.3 | 2.0 | **81.3** |

performed as follows:

$$x_s \in \mathbb{R}^{B \times C_t \times H_t \times W_t} \rightarrow \hat{x}_s \in \mathbb{R}^{C_t \times BN_t},$$

$$\mu_s = \mathrm{mean}(\hat{x}_s), \quad \sigma_s = \mathrm{std}(\hat{x}_s),$$

$$\hat{x}_s = \frac{\hat{x}_s - \mu_s}{\sigma_s} \cdot \sigma_t + \mu_t$$

where $s$ denotes FusionFormer and $t$ denotes ViT, $\mathrm{mean}$ and $\mathrm{std}$ are opreated over the last dimension. To enable high-level semantic guidance, a compact global vector is extracted from $\hat{x}_s$ through global average pooling. This vector is then aligned with the VFM's class token using an MSE loss, inspired by the work of Zhang et al. (2025).

## 3 EXPERIMENTS

**Dataset.** We pretrain FusionFormer on the ImageNet-1K classification benchmark Deng et al. (2009); Russakovsky et al. (2015) to assess its representational capacity as a versatile and efficient backbone. The pretrained model is then used to initialize FusionFormer for segmentation tasks. We evaluate the model on two widely-used semantic segmentation benchmarks: ADE20K Zhou et al. (2017; 2019) and Cityscapes Cordts et al. (2016; 2015). ADE20K contains 20,210 training images, 2,000 validation images, and 3,352 test images annotated with 150 fine-grained semantic categories. Cityscapes focuses on high-resolution urban street scenes with 19 semantic classes, including 2,975 finely annotated training images and 500 validation images.

**Implementation Details.** We follow the standard ImageNet training protocol of Swin Transformer Liu et al. (2021) and report Top-1 accuracy on the validation set. For segmentation tasks, our implementation is based on MMSegmentation Contributors (2020), and we apply the same data augmentations from prior works Wan et al. (2023); Xu et al. (2024); Cao et al. (2024); Yan et al. (2024). Among recent VFMs, we adopt DINOv2 Oquab et al. (2023) as the semantic guidance source, due to its SOTA performance and strong generalization ability across dense prediction tasks. Additional training and implementation details are provided in Appendix A.1.

Table 2: Comparisons with SOTA real-time segmentation methods on the Cityscapes and ADE20K validation sets. Parameters (Params) are in Millions (M) and FLOPs are in GigaFLOPs. Both FLOPs and FPS are reported at input resolutions of $2048 \times 1024$ for Cityscapes and $512 \times 512$ for ADE20K.

| Method | Params | Cityscapes | | | ADE20K | | |
|---|---|---|---|---|---|---|---|
| | | FLOPs | FPS | mIoU | FLOPs | FPS | mIoU |
| SegFormer-B0 Xie et al. (2021) | 3.8 | 125.5 | 30.2 | 76.2 | 8.4 | 84.4 | 37.4 |
| SeaFormer-S Wan et al. (2023) | 4.0 | 8.0 | 50.5 | 76.1 | 1.1 | 67.5 | 38.1 |
| VWFormer-B0 Yan et al. (2024) | 3.7 | – | – | 77.2 | 5.8 | – | 38.9 |
| SDPT-Tiny Cao et al. (2024) | 3.6 | 63.4 | 47.3 | 77.3 | 5.7 | 63.8 | 39.4 |
| SeaFormer-S++ Wan et al. (2025) | 4.1 | 8.0 | 50.5 | 77.2 | 1.1 | 67.5 | 39.7 |
| FusionFormer-S | 3.6 | 26.5 | **75.4** | **78.2** | 3.4 | **159.9** | **40.8** |
| TopFormer-B Zhang et al. (2022b) | 5.1 | 11.2 | 81.4 | 75.0 | 1.8 | 96.2 | 37.8 |
| DDRNet-23-S Pan et al. (2022a) | 5.7 | 36.3 | 106.7 | 77.8 | 18.2 | – | 33.3 |
| SegNext-T Guo et al. (2022) | 4.3 | 50.5 | 28.1 | 79.8 | 6.6 | 60.3 | 41.1 |
| PIDNet-S Xu et al. (2023) | 7.6 | 67.6 | 93.2 | 78.8 | – | – | – |
| SeaFormer-B Wan et al. (2023) | 8.6 | 13.7 | 37.5 | 77.7 | 1.8 | 44.5 | 40.2 |
| SeaFormer-B++ Wan et al. (2025) | 8.6 | 13.7 | 37.5 | 78.6 | 1.8 | 44.5 | 41.4 |
| FusionFormer-B | 7.7 | 54.3 | 53.2 | **80.2** | 6.9 | **127.9** | **45.3** |
| SegFormer-B1 Xie et al. (2021) | 13.7 | 243.7 | 22.5 | 78.5 | 15.9 | 55.8 | 42.2 |
| HRFormer-S Yuan et al. (2021) | 13.5 | 835.7 | – | 80.0 | 109.5 | – | 44.0 |
| RTFormer-B Wang et al. (2022a) | 16.8 | – | 50.2 | 79.3 | 67.4 | 93.4 | 42.1 |
| DDRNet-23 Pan et al. (2022a) | 20.1 | 143.1 | 45.7 | 79.5 | 71.6 | 134.1 | 38.8 |
| PIDNet-M Xu et al. (2023) | 34.4 | 197.4 | 39.8 | 80.1 | – | – | – |
| SeaFormer-L Wan et al. (2023) | 14.0 | 50.2 | 28.5 | 79.4 | 6.5 | 37.8 | 42.7 |
| SCTNet-B Xu et al. (2024) | 17.4 | 119.8 | **62.8** | 80.5 | 15.0 | **145.1** | 43.0 |
| VWFormer-B1 Yan et al. (2024) | 13.7 | – | – | 79.0 | 13.2 | – | 43.2 |
| SDPT-Small Cao et al. (2024) | 11.9 | 131.3 | 32.8 | 80.4 | 12.7 | 46.8 | 46.0 |
| FusionFormer-L | 13.1 | 89.2 | 43.4 | **80.9** | 11.3 | 110.6 | **46.7** |

## 3.1 PERFORMANCE ON IMAGENET-1K

ImageNet-1K serves as a fundamental benchmark to evaluate the representational capacity of visual backbones. For the input size of $224 \times 224$, we configure the window sizes and fusion ratios as $\{(7, 1), (14, 2)\}$ and $\{(7, 1)\}$ for the last two stages. As shown in Table 1, FusionFormer consistently outperforms recent SOTA lightweight models under comparable computational budgets, achieving superior Top-1 accuracy. In particular, FusionFormer-B achieves a noticeable improvement over SeaFormer-B Wan et al. (2023) and MiT-B1 Xie et al. (2021), demonstrating the effectiveness of MWFA. These results verify the potential of FusionFormer as a versatile and competitive backbone.

## 3.2 COMPARISON WITH SOTA SEGMENTATION METHODS

We compare our method with SOTA real-time methods on two widely-used benchmarks, Cityscapes and ADE20K, with mIoU as the accuracy metric and FLOPs as the efficiency metric.

**Results on Cityscapes.** As shown in Table 2 (left), FusionFormer achieves SOTA performance on the Cityscapes dataset Cordts et al. (2016), while maintaining an excellent trade-off between accuracy and computational cost. For instance, FusionFormer-L yields an $80.9\%$ mIoU, surpassing SCTNet-B Xu et al. (2024) ($80.5\%$ mIoU) by $0.4\%$ mIoU while utilizing only $74\%$ of its computational budget. Notably, FusionFormer-B establishes a new SOTA record at its scale with $80.2\%$ mIoU, significantly outperforming prior methods, even larger models like SegFormer-B1 Xie et al. (2021) and SeaFormer-L Wan et al. (2023). FusionFormer-S also delivers competitive results, underscoring the robustness of our design across model scales. Interestingly, employing semantic guidance with the class token leads to a slight performance degradation compared to direct training. We attribute this to the challenges of modeling global dependencies for high-resolution inputs, which can render such semantic guidance less effective. This observed limitation underscores the advantage of our proposed MWFA in efficiently modeling diverse dependencies across both local and long-range regions. More ablations are detailed in Section 3.3

Table 3: Ablation study of MWFA components. We evaluate the effect of multi-window setting (MW), fusion weights ($\mathcal{F}$), head-aware fusion ($\mathcal{F}_r$), and window-level weighting ($\alpha_i$).

| MW | $\mathcal{F}$ | $\mathcal{F}_r$ | $\alpha_i$ | Params | FLOPs | FPS | mIoU |
|----|----|----|----|----|----|----|----|
| – | – | – | – | 7.4M | 6.88G | 144.8 | 42.6 |
| ✓ | – | – | – | 7.4M | 6.64G | 142.5 | 42.9 |
| ✓ | ✓ | – | – | 7.7M | 6.89G | 134.0 | 44.0 |
| ✓ | – | – | ✓ | 7.4M | 6.64G | 139.1 | 42.8 |
| ✓ | ✓ | ✓ | – | 7.7M | 6.89G | 132.9 | 43.6 |
| ✓ | ✓ | – | ✓ | 7.7M | 6.89G | 128.1 | 43.7 |
| ✓ | ✓ | ✓ | ✓ | 7.7M | 6.89G | 127.9 | **44.2** |

Table 4: **Left:** Impact of different guidance source. **Right:** Ablation of different efficient attention.

| Patch tokens | Class token | Cityscapes mIoU | ADE20K mIoU |
|----|----|----|----|
| – | – | 80.0 | 44.2 |
| ✓ | – | **80.2** | 44.5 |
| – | ✓ | 79.5 | 44.9 |
| ✓ | ✓ | 79.2 | **45.3** |

| Attention type | Source model | ADE20K mIoU |
|----|----|----|
| Window | Without MW | 45.2 |
| Axial | SeaFormer-L | 44.9 |
| Conv | SCTNet-B | 44.0 |
| MWFA | Ours-L | **46.7** |

**Results on ADE20K.** The results on ADE20K Zhou et al. (2017) are shown in Table 2 right. With the guidance from VFMs, FusionFormers consistently deliver SOTA performances. For instance, FusionFormer-L achieves 46.7% mIoU with only 11.3 GFLOPs, outperforming the previous best model, SDPT-Small Cao et al. (2024), by 0.7% mIoU. FusionFormer-B continues to exhibit strong performance, achieving 45.3% mIoU, outperforming all models at a similar scale and even all others except SDPT-Small. On the ADE20K dataset with greater category diversity and potentially a smaller input size that benefits guidance efficacy, VFM guidance yields a more pronounced performance improvement. This substantial gain demonstrates that such guidance enhances both fine-grained detail capture and overall accuracy, proving particularly effective in multi-category scenes.

## 3.3 ABLATION STUDY

**Impact of components in MWFA.** We conduct a detailed ablation study on ADE20K to evaluate the contribution of each component in our proposed MWFA. As shown in Table 3, the baseline without any MWFA module corresponds to the standard Swin block, which achieves 42.6% mIoU. Incorporating the multi-window (MW) configuration with local region fusion yields a minor gain to 42.9% mIoU and slightly reduces FLOPs. Adding fusion weights ($\mathcal{F}$) brings a notable gain to 44.0%, indicating that adaptive local token aggregation is more effective than simple average within fixed windows. We further explore head-aware fusion ($\mathcal{F}_r$) and window-level weighting ($\alpha_i$), both of which contribute independently to performance (43.6% and 43.7% mIoU, respectively) when added on top of MW and $\mathcal{F}$. Combining all components yields the best result of 44.2% mIoU, validating the effectiveness and complementarity of our MWFA design in enhancing dense prediction performance while maintaining lightweight computation.

**The influence of different guidance source from foundation models.** Table 4 (**left**) summarizes our findings. For high-resolution Cityscapes, demanding precise localization, fine-grained patch tokens slightly improved mIoU (to 80.2% from a baseline of 80.0%). Conversely, the class token was detrimental, reducing performance whether used alone (79.5% mIoU) or when combined with patch tokens (79.2% mIoU), suggesting that generalized global information can be unsuitable. In contrast, for the diverse ADE20K dataset, both token types proved advantageous. Patch tokens offered a modest gain (44.5% mIoU from 44.2%), while class tokens provided a stronger individual boost (44.9% mIoU), and their combination achieved the top performance (45.3% mIoU). These findings underscore the need to tailor VFM guidance strategies to specific task characteristics.

**Evaluating Efficient Attention Mechanisms for ViT-Guided Segmentation.** We compare the utilization of guidance from mainstream efficient attention mechanisms on the ADE20K dataset, where VFMs have shown significant improvements and excellent performance. As detailed in Table 4 (Right), our MWFA (from FusionFormer-L) achieves a leading 46.7% mIoU. This performance

markedly surpasses other efficient attentions: Window Attention Liu et al. (2021) (from 'Without MW' configuration, 45.2%), Axial Attention (based on SeaFormer-L Wan et al. (2023), 44.9%), and Conv Attention (based on SCTNet-B Xu et al. (2024), 44.0%). The results, showing an improvement of 1.5 to 2.7% mIoU , affirm MWFA's superior efficacy in leveraging contextual information for enhanced semantic segmentation.

## 4 RELATED WORK

**Real-time Segmentation Methods.** Real-time semantic segmentation requires a balanced trade-off between accuracy and efficiency. HRFormer Yuan et al. (2021) incorporates window-based self-attention into the multi-resolution stream of HRNet Wang et al. (2020b), maintaining strong spatial representations with manageable complexity. TopFormer Zhang et al. (2022b) pushes efficiency further by confining global self-attention to the 1/64 scale. RTFormer Wang et al. (2022a) achieves a better trade-off by applying self-attention at intermediate resolutions (1/16 and 1/32), and introducing a Cross-Resolution Attention mechanism that allows high-resolution features to aggregate long-range features from pooled low-resolution features. SeaFormer Wan et al. (2023; 2025) adopts an enhanced axial attention mechanism Ho et al. (2019); Wang et al. (2020a), and simplifies feature fusion through convolution and element-wise operations, achieving a leading efficiency. Despite these improvements, the constrained receptive fields often limit their ability to capture long-range context. SCTNet Xu et al. (2024) introduces a training-only Transformer branch based on SegFormer Xie et al. (2021), which provides strong long-range semantic priors. Meanwhile, the lightweight path adopts convolutional attention Chen et al. (2020); Guo et al. (2022) to effectively absorb semantic and spatial cues guided by the SegFormer branch. However, such methods often rely on guidance provided by larger models employed within task-specific frameworks, such as SeaFormer++ Wan et al. (2025) which uses larger SeaFormer variants for feature guidance.

**Efficient Attention Mechanisms.** Attention mechanisms have demonstrated strong performance in various vision tasks, but the quadratic complexity limits their deployment in real-time scenarios. Several strategies have been proposed to address this limitation. Region-restricted methods, such as Window Attention Huang et al. (2019a); Vaswani et al. (2021), confine attention computation within non-overlapping windows. Further improvements introduce shifting mechanisms Liu et al. (2021; 2022) to enable cross-window interactions. However, the limited receptive field of small windows requires deep stacking to approximate long-range modeling. Token compression approaches, such as Axial Attention Huang et al. (2019b); Ho et al. (2019); Wang et al. (2020a); Wan et al. (2023), reduce computational cost by separately attending along height and width axes, or by applying local pooling operations Cao et al. (2024); Norouzi et al. (2024). Yet, axial decomposition hinders inter-axis interactions, and local pooling lacks the capacity for linear-complexity global context modeling. Token selection methods, such as Deformable Attention Zhu et al. (2020), significantly reduce the number of attended tokens by learning to sample a fixed set of key positions globally. Alternatively, some designs select row- or column-wise tokens based on structural priors without introducing additional parameters Su et al. (2024a). Nevertheless, global token selection may result in inconsistency Zhang et al. (2023) due to the learned sampling patterns.

## 5 CONCLUSION

In this paper, we introduce FusionFormer, a lightweight Transformer incorporating MWFA, specifically designed for efficient local-to-global context modeling. Our MWFA enables effective leverage of global guidance from VFMs, integrating semantic priors to boost performance. Moreover, we present a family of FusionFormer variants and achieve an excellent accuracy-efficiency balance. FusionFormer's results on ImageNet-1K classification demonstrate its potential as a versatile and efficient vision backbone. Subsequently, its superior performance on the ADE20K and Cityscapes segmentation benchmarks further validates the effectiveness of our VFM-guided approach.

**Limitations and Broader Impacts.** Our work enables the efficient deployment of VFMs, democratizing AI in data-scarce domains like medical imaging Chen et al. (2024); Vorontsov et al. (2024), and we leave visual-language model integration for future work. However, overconfidence in our method may introduce risks in scenarios with drastic scene changes or severe data scarcity, as achieving robust data efficiency under these challenging conditions remains an ongoing research endeavor.

ETHICS STATEMENT

Our research on efficient real-time semantic segmentation is primarily intended for positive societal applications, such as enhancing perception systems in autonomous driving, robotics, and medical imaging analysis. We have conducted our research using publicly available, standard academic datasets (ImageNet-1K, Cityscapes, and ADE20K), which have established protocols for data privacy and usage.

REPRODUCIBILITY STATEMENT

We are committed to ensuring the reproducibility of our research. To facilitate this, we provide the following resources and details:

**Source Code and Models:** To facilitate reproducibility, we have included our source code in the supplementary material. The core implementation of our FusionFormer backbone is located at `./mmseg/models/backbones/fusionformer.py`, and the provided codebase contains all necessary scripts to train our models. Upon acceptance, we will publicly release the complete codebase for the entire experimental workflow, along with all pre-trained model weights, in a public repository.

**Architecture and Configurations:** The detailed architecture of our FusionFormer, including the design of the STEM and the Multi-Window Fusion Attention (MWFA) mechanism, is described in Section 2. Specific configurations for each model variant (FusionFormer-S, -B, -L), including embedding dimensions and attention head parameters, are provided in Appendix A.1.

**Training and Hyperparameters:** All hyperparameters and training procedures for both ImageNet-1K pretraining and downstream semantic segmentation tasks on Cityscapes and ADE20K are detailed in Section 3 and Appendix A.1. This includes optimizer settings, learning rate schedules, data augmentations, and the specific configurations for our VFM guidance mechanism.

**Datasets:** All datasets used in this work (ImageNet-1K, Cityscapes, ADE20K) are standard public benchmarks, with references provided in the main text. The DINOv2-B/14 model used for guidance is the official version released by its authors.

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

# A APPENDIX

## A.1 MORE IMPLEMENTATION DETAILS

**ImageNet Pre-training.** For ImageNet pre-training, we follow the standard training protocol of Swin Transformer Liu et al. (2021) on 4 NVIDIA RTX 4090 GPUs. The main details of the pre-training settings are as follows, where MW settings denotes the window sizes and fusion ratios for the last two stages:

Table 5: Training settings on ImageNet-1K.

| config | value |
|---|---|
| optimizer | AdamW |
| base learning rate | 0.001 |
| weight decay | 0.05 |
| optimizer momentum | $\beta_1, \beta_2 = 0.9, 0.999$ |
| learning rate schedule | cosine decay |
| minimum learning rate | 1e-5 |
| warmup epochs | 20 |
| warmup learning rate | 1e-6 |
| training epochs | 300 |
| batch size | 1024 |
| augmentation | RandAug(9, 0.5) |
| random resized crop | 224 |
| random flip | 0.5 |
| mixup | 0.8 |
| cutmix | 1.0 |
| random erasing | 0.25 |
| label smoothing | 0.1 |
| MW settings | [(7, 1), (14, 2)], [(7, 1)] |

**Segmentation Training.** For all semantic segmentation tasks, we train all models using the AdamW optimizer Loshchilov & Hutter (2017) with an initial learning rate of 0.00006 and a weight decay of 0.05. We employ a polynomial learning rate decay policy (power 1.0) to adjust the learning rate. Standard data augmentation methods are applied, including random cropping, random scaling (from 0.5 to 2.0), and random horizontal flipping. All models are trained for 160k iterations. The batch size is set to 16 for Cityscapes and 32 for ADE20K. The guidance process utilizes the same core training settings (e.g., optimizer, learning rate policy, iteration count) and data augmentation techniques as described above. However, the specific VFM guidance signals are adapted based on dataset characteristics: for the high-resolution Cityscapes dataset, we exclusively employ patch tokens, whereas for ADE20K, both patch and class tokens are utilized.

**Inference and Evaluation Details.** To evaluate the real-time performance of our models, all inference speed (FPS) benchmarks were conducted on a single NVIDIA RTX 3090 GPU. Our Fusion-Former is implemented entirely with standard PyTorch modules, without reliance on customized CUDA operators often found in libraries like MMCV. This design choice makes our model highly compatible with modern compiler optimizations such as `torch.compile` (available in PyTorch 2.0 and later). Therefore, during FPS benchmarking, we applied `torch.compile` to our model to ensure that we are measuring its true inference performance in an efficient, production-like deployment setting.

**Model Configurations.** We derive three model variants: FusionFormer-Small (S), FusionFormer-Base (B), and FusionFormer-Large (L) for fair comparisons with previous works. We unify the number of attention heads to 16, and more specific configurations are as follows:

- **FusionFormer-S:** Initial dim $C_0 = 32$; MWFA head dim $= \{24, 24\}$.
- **FusionFormer-B:** Initial dim $C_0 = 48$; MWFA head dim $= \{32, 32\}$.
- **FusionFormer-L:** Initial dim $C_0 = 64$; MWFA head dim $= \{32, 48\}$.

The $(w, r)$ pairs for our MWFA are specifically configured for each dataset in the final two stages of FusionFormer, due to their different resolutions. On the Cityscapes dataset, the penultimate MWFA

stage employs the configuration $(8, 2), (16, 2), (32, 4)$. On the ADE20K dataset, these configurations are $(8, 2), (16, 2)$ for the penultimate stage. As we employ fixed number of blocks with $6, 2$ for the last two stages, these blocks alternate between MWFA and SMWFA, in which the windows are cyclically shifted by half of their respective sizes.

**Selection of Fusion Ratios.** The selection of the fusion ratio $r$ for each window size $w$ in MWFA is guided by a key principle: the total computational cost must remain lower than that of an efficient baseline (e.g., standard window attention with $w = 8$), while simultaneously avoiding excessive fusion that could harm detail preservation. We use this principle to determine the settings for our high-resolution stages. For instance, for a stage with a multi-window configuration of $w = \{8, 16, 32\}$, the attention complexity of a standard $w = 8$ window attention baseline is approximately $2Nd \cdot 64$. We set the corresponding fusion ratios to $r = \{2, 2, 4\}$, which results in a total attention complexity for MWFA of: $2Nd \cdot (\frac{8^2}{2^4} + \frac{16^2}{2^4} + \frac{32^2}{4^4}) = 2Nd \cdot (4 + 16 + 4) = 2Nd \cdot 24$. This configuration is therefore substantially more efficient than the baseline, yet approximates a much larger receptive field.

## A.2 MORE PERFORMANCE DETAILS

This section provides additional performance details for our FusionFormer variants. We present a comparative evaluation of their semantic segmentation capabilities on the Cityscapes and ADE20K benchmarks, contrasting results from models trained with standard ImageNet pretraining against those augmented with guidance from Vision Foundation Models.

Table 6: Performance comparison of FusionFormer variants with and without VFM guidance on Cityscapes and ADE20K. All results are reported as mIoU (%). Symbols indicate VFM guidance: ✗ (none), ✓ (applied).

| Model Variant | VFM Guidance | Cityscapes(%) | ADE20K(%) |
|---|:---:|:---:|:---:|
| FusionFormer-S | ✗ | **78.2** | 39.9 |
| | ✓ | 78.0 | **40.8** |
| FusionFormer-B | ✗ | 80.0 | 44.2 |
| | ✓ | **80.2** | **45.3** |
| FusionFormer-L | ✗ | 80.6 | 45.5 |
| | ✓ | **80.9** | **46.7** |

The performance impact of incorporating VFM guidance is detailed in Table 6, comparing against a baseline of ImageNet pretraining only for our FusionFormer variants (Small, Base, and Large) on Cityscapes and ADE20K. Overall, the introduction of VFM guidance tends to enhance segmentation accuracy, particularly for the larger model variants.

Specifically, for FusionFormer-B and FusionFormer-L, VFM guidance consistently yields superior mIoU scores on both datasets. For instance, FusionFormer-L with VFM guidance achieves 80.9% mIoU on Cityscapes (an improvement from 80.6%) and a notable 46.7% mIoU on ADE20K (a gain of 1.2% over its baseline of 45.5%). Similarly, FusionFormer-B benefits across the board, with mIoU increasing from 80.0% to 80.2% on Cityscapes and from 44.2% to 45.3% on ADE20K when VFM guidance is applied.

An interesting observation arises with the FusionFormer-S variant. While VFM guidance substantially boosts its performance on ADE20K, increasing the mIoU from 39.9% to 40.8%, it results in a slight performance decrease on Cityscapes (78.0% with guidance versus 78.2% without). This nuanced outcome suggests that for highly compact models like FusionFormer-S, the benefits of VFM guidance might be more dataset-dependent, potentially requiring specific tuning or alternative integration strategies for optimal performance on high-resolution datasets like Cityscapes.

## A.3 COMPLEXITY ANALYSIS

To evaluate the computational cost of our Multi-Window Fusion Attention (MWFA), we will estimate its floating-point operations (FLOPs). Our estimation is grounded in the foundational un-

derstanding of matrix multiplication complexity: specifically, multiplying two matrices of shapes $\mathbb{R}^{N \times C}$ and $\mathbb{R}^{C \times M}$ to produce an $\mathbb{R}^{N \times M}$ matrix involves $NMC$ multiplications and $NM(C-1)$ additions, leading to approximately $2NMC$ FLOPs.

Given MWFA with a window size of $w$ and a fusion ratio of $r$, we can derive the FLOPs for the attention mechanism as follows:

$$\text{FLOPs} = \frac{2N}{w^2}(\frac{w^2}{r^2})^2 d = \frac{2Nw^2d}{r^4}$$

where $N$ is the number of tokens, $d$ is the dimension. The term $\frac{w^2}{r^2}$ represents the number of tokens in each fused window.

We further extend it to MWFA with multiple windows and fusion ratios. For a stage with $K$ windows and $K$ fusion ratios, the total FLOPs can be expressed as:

$$\text{FLOPs} = 2Nd \sum_{i=1}^{K} \frac{w_i^2}{r_i^4}$$

Therefore, the computational complexity of MWFA is linear with respect to the number of tokens $N$, i.e., $\mathcal{O}(Nd)$.

For our FusionFormer-B variant, when processing high-resolution 2048×1024 inputs characteristic of the Cityscapes dataset, the total computational cost is 55.104 GFLOPs. This budget is primarily distributed across its main components as follows: the initial feature extraction stem accounts for 16.694 GFLOPs, the core Multi-Window Fusion Attention (MWFA) stages contribute a significant 34.974 GFLOPs, and the lightweight segmentation head adds 3.431 GFLOPs. Notably, the MWFA stages represent the largest portion of the computation (approximately 63.5% of the total FLOPs), underscoring the critical importance of their efficient design to the overall efficiency.

### A.4    ABLATION STUDY ON POSITION EMBEDDING

We conduct an ablation study to validate our choice of position embedding for the Multi-Window Fusion Attention (MWFA) architecture, as some position encoding methods are not ideally suited for our multi-window design. While Relative Position Bias (RPB) is effective for standard fixed-size window attention, adopting it in our context would require learning a separate, non-shareable set of biases for each window size (e.g., $w_i = \{8, 16, 32\}$), significantly increasing complexity. Moreover, its intra-window nature lacks the cross-window spatial awareness we aimed for. We therefore compare our chosen approach, 2D Rotary Position Embedding (RoPE), against RPB and a unified Absolute Positional Embedding (APE) baseline.

The results for our FusionFormer-B model (trained without VFM guidance) are presented in Table 7. Our choice of 2D RoPE is empirically validated, as it consistently outperforms both APE and the multi-set RPB on both datasets. While relative encodings (RPB and 2D RoPE) proved superior for the diverse scenes in ADE20K, the simpler APE surpassed the complex multi-set RPB on the highly structured Cityscapes dataset. This highlights the robustness of 2D RoPE's unified spatial context, making it the optimal and most effective choice for our multi-window architecture across different data distributions.

Table 7: Ablation study on different position embedding methods. The study is conducted on FusionFormer-B (trained from scratch without VFM guidance) on the ADE20K and Cityscapes validation sets. Results are reported in mIoU (%). Best results are in **bold**.

| Dataset    | APE  | RPB  | 2D RoPE (Ours) |
| ---------- | ---- | ---- | -------------- |
| ADE20K     | 43.3 | 43.8 | **44.2**       |
| Cityscapes | 79.5 | 79.2 | **80.0**       |

### A.5    LARGE LANGUAGE MODEL (LLM) USAGE STATEMENT

In accordance with the ICLR 2026 submission guidelines, we disclose the use of large language models (LLMs) in the preparation of this paper. We employed ChatGPT (an OpenAI language

model) solely to aid in polishing the writing and improving readability. The LLM was not used for research ideation, data analysis, code generation, or producing original scientific content. All technical contributions, experimental designs, analyses, and conclusions are entirely the work of the authors. The authors take full responsibility for the content of this paper.

