# OpenReview forum: "FusionFormer: Multi-Window Fusion for Efficient Real-Time Segmentation with Vision Foundation Models"
_ICLR.cc/2026/Conference — Submitted to ICLR 2026_

### Official Review · Reviewer_UZkc · 2025-10-24

**Soundness:** 3
**Presentation:** 3
**Contribution:** 2
**Rating:** 6
**Confidence:** 3

**Summary:**

In this work the authors propose a backbone architecture and training framework for efficient semantic segmentation.
The authors recognize the speed/accuracy tradeoff in windowed attention, where increases in window size improve global understanding while increasing computational complexity of attention mechanisms quadratically.
FusionFormer is designed to approximate global contextual dependencies while maintaining low computational complexity.

The main architectural novelty is the Multi-Window Fusion Attention (MWFA) mechanism.
MWFA addresses the quadratic complexity of global attention by restricting computations to local windows and fusing tokens adaptively.
It captures interactions from short-range (local details) to long-range (global context) using multiple window sizes and fusion ratios.
MWFA projects QKV using linear layers, in a way that is identical to standard self-attention.

For a given window configuration (window size w), the projected QKV tensors are split into non-overlapping windows.
In order to reduce computation in larger windows (note: computational complexity is a function of w^2), an adaptive token fusion mechanism is used: a convolution-based MLP is applied to the concatenated QKV to generate fusion weights.
Tokens are fused by groups of r^2 (r: fusion ratio), weighed by the fusion weights.
In order to promote diversity in fused features, a learnable per-head bias term is added in order to bias the weights.
MWFA combines multiple configurations of window sizes and fusion ratios by weighing them based on a heuristic approximation of confidence, using attention scores.

Another novelty of the paper is the VFM guidance, which enables knowledge distillation from (potentially much less efficient) backbones, such as DINOv2.

The overall backbone architecture consists of a convolutional STEM for strides 4/8/16, two MWFA stages at stride 16, alternating MWFA and (shifted) SMWFA, and a final stride-32 convolution block.

Experiments are showing competitive results on the speed/accuracy Pareto frontier. Experiments include results for image classification (ImageNet-1k) and Semantic Segmentation (ADE20k and Cityscapes).

**Strengths:**

The paper is very well written, and the motivation is clear.

The idea of mixing window sizes and fusion ratios is interesting and seems novel.

The distillation framework from Vision Foundation Models, although not ground breaking, is a strong addition to the method.

**Weaknesses:**

The baselines that are used for comparison are old.

Prior works also aim at addressing the tradeoff of local/global attention:
* EdgeViT (ECCV2022) decomposes self-attention into convolution-based local aggregation and sparse global attention on delegate tokens.
* FasterViT (ICLR2024) employs local-to-global attention with Hierarchical Attention (carrier tokens) and appears to show competitive results, although they are not easy to compare against FusionFormer due to the different compute (FLOPS) budgets.

For VFMs, RADIO2.5 (CVPR 2025) improves upon DINOv2 on semantic segmentation (linear probe), and DINOv3 further improves from there (DINOv3 is, to my knowledge, a pre-print, not yet published in a conference, thus the authors are excused for not mentioning it).

**Questions:**

Did you ablate the need for the final MBConv block used for downsampling to a stride of 32? It would seem like semantic segmentation could benefit from a smaller patch size, thus why not stick to a patch size of 16?

Did you try to use a more expressive (e.g. UperNet) semantic segmentation head?

---

> ### Author Response · Authors · 2025-11-15
> **Our rebuttal to reviewer UZkc**
>
> Thank you for your in-depth review and meticulous summary of our paper. Your expert feedback is invaluable for improving our work, and we will address your questions and concerns one by one.
>
> # Main Comments
> > **Q1 (Weakness 1): The baselines that are used for comparison are old.**
>
> **A1:** We appreciate you raising this critical point and highlighting relevant SOTA works. We acknowledge that our initial draft primarily focused on comparing against models within our direct competitive niche, which includes models like SCTNet (AAAI 2024), SDPT (TITS 2024), and SeaFormer++ (IJCV 2025).
>
> Regarding the specific works you mentioned:
>
> - **EdgeViT (ECCV 2022):** We agree that EdgeViT is an important work in efficient hybrid attention. However, it does not provide official, directly comparable results on standard real-time segmentation benchmarks (e.g., Cityscapes, ADE20K), which makes it difficult to include as a direct performance baseline.
>
> - **FasterViT (ICLR 2024):** While FasterViT is a powerful model, its design targets a different scale and scope. Its smallest variant, FasterViT-0, already has 31.4M parameters, which is substantially larger than our largest model, FusionFormer-L (13.1M). In terms of both parameter count and computational complexity, FasterViT operates in a different weight class and is not directly comparable to models in the lightweight, real-time domain we focus on.
>
> Nevertheless, we sincerely appreciate your valuable suggestions. To provide a more comprehensive overview of the field for our readers, we will expand the **Efficient Attention Mechanisms** part in Section **Related Work** of our final manuscript. We will incorporate a discussion of these works there, analyzing their distinct strategies for tackling the local/global attention trade-off.
>
> > **Q2 (Weakness 2): For VFMs, RADIO2.5 (CVPR 2025) improves upon DINOv2 on semantic segmentation (linear probe), and DINOv3 further improves from there.**
>
> **A2:** Thank you for highlighting these recent advancements in VFMs. Regarding the specific models you mentioned:
>
> - **RADIOv2.5 (CVPR 2025):** We note that the core contribution of RADIOv2.5 lies in aggregating knowledge from multiple expert teachers (e.g., DINOv2, SAM) into a single student model via **multi-teacher distillation**. Its design goals and model scale differ significantly from our work. On ADE20k task, its smallest variant, **RADIOv2.5-B**, has 98M parameters and achieves 48.9% mIoU.
>
> - **DINOv3:** We acknowledge that DINOv3 has made significant strides over DINOv2 by refining the loss function and leveraging larger-scale data. Our choice of DINOv2 as the guidance source was based on its status as one of the most extensively validated open-source VFMs available during the main development phase.
>
> We wish to emphasize that a core contribution of our framework is its ability to efficiently leverage guidance from any powerful VFM within a lightweight segmentation architecture. Our framework is, by design, model-agnostic. We are confident that replacing the guidance source with DINOv3 would further boost our model's performance. However, at the time of writing, we did not have access to a publicly available, smaller-scale pre-trained DINOv3 model that would be suitable for our experimental setup.
>
> > **Q3: the final MBConv block used for downsampling to a stride of 32.**
>
> **A3:** This is an excellent question that addresses a key trade-off between efficiency and performance in our architectural design. We opted to include the final MBConv block for two primary reasons:
>
> - **Maintaining Consistent Complexity for Fair Comparison:** To ensure fair comparisons, our overall architecture, in terms of stage division and downsampling strategy, largely follows the conventions established by mainstream real-time segmentation methods like SCTNet and SDPT, which typically downsample to a stride of 32 in their final stages. Adhering to this convention allows us to innovate upon a comparable baseline of computational complexity.
>
> - **Mitigating the Computational Burden of High-Resolution Inputs:** This is particularly critical when processing high-resolution tasks such as Cityscapes. With an input resolution of $2048 \times 1024$, the feature map at a stride of 32 still contains $64 \times 32 = 2048$ visual tokens, which represents a substantial computational load. If we were to maintain a stride of 16, the number of tokens would quadruple, leading to an immense increase in the computational and memory footprint during the decoding stage. This would severely compromise the model's real-time performance.
>
> We acknowledge, as you pointed out, that maintaining a stride of 16 in the final two stages could be more beneficial for feature alignment with VFM patch tokens (which are often at stride 14 or 16). However, our final design choice was a deliberate trade-off, prioritizing strict control over computational complexity and the need for fair comparisons with prior art.

---

> > ### Author Response · Authors · 2025-11-15
> > **Our rebuttal to reviewer UZkc (continued)**
> >
> > Due to the character limit of the comment box, we will post our response to the final question in a separate comment immediately following this one.
> >
> > > **Q4: Did you try to use a more expressive (e.g. UperNet) semantic segmentation head?**
> >
> > **A4:** Thank you for this excellent suggestion. Our choice of a minimalist, lightweight head was primarily driven by the goal of ensuring a **fair and consistent comparison** with current SOTA methods.
> >
> > We observed that the leading models we benchmark against, such as **SCTNet** and **SeaFormer**, predominantly employ lightweight decoding strategies. In fact, to maximize fairness in our comparison, the **Light Head** we adopted is directly inspired by the lightweight decoding design used in SeaFormer.
> >
> > Introducing a novel and more complex head like **UperNet** into our framework would create an **unfair comparative advantage**. It would make it difficult to ascertain whether performance gains stem from our innovative **FusionFormer backbone** or from the powerful capabilities of the head itself. Such a choice could potentially lead to a misattribution of our core contribution's value.
> >
> > Therefore, by adopting a decoder design similar to that of a key competitor, we can effectively isolate the variables. This allows us to more clearly demonstrate that the observed performance improvements genuinely originate from the capabilities of our FusionFormer backbone. Our results in Table 2 validate that even in this head-to-head setting, our backbone is powerful enough to drive the model to achieve SOTA performance.
> >
> > Once again, we sincerely thank you for your constructive comments and insightful feedback. Your review has been immensely helpful in refining our work. We hope that our responses have thoroughly addressed all of your concerns. We are confident that by incorporating your valuable suggestions, the final version of our paper will be significantly strengthened.

---

> > ### Comment · Reviewer_UZkc · 2025-11-15
> >
> > Thank you for posting your responses. I have no major concern about the paper, I think it is valuable and well researched. However the proposed approach seems incremental, and doesn't introduce radically new ideas. For this reason, I would stick to my current assessment.

---

### Official Review · Reviewer_2wTm · 2025-11-01

**Soundness:** 3
**Presentation:** 3
**Contribution:** 3
**Rating:** 6
**Confidence:** 3

**Summary:**

The paper proposes FusionFormer, an efficient real-time segmentation foundation model that features the novel multi-window fusion transformer layers. In multi-window fusion transformer, the computational complexity is linear by constraining self-attention within the window. The experiment is carried out on ImageNet-1K dataset and ADE20K and Cityscape segmentation benchmarks. The proposed method is able to reach a balance between computational cost and segmentation accuracy.

**Strengths:**

- The multi-window fusion transformer can capture dependencies from short to long-range interactions, which is a significant improvement over traditional window attention.
- The proposed method improves the FPS in segmentation benchmarks, realizing real-time processing.

**Weaknesses:**

- The efficiency of FusionFormer-L is no as good as other model settings.
- The proposed method is mainly demonstrated in segmentation task, the reviewer is wondering if the similar architecture can be extended to be used in foudation models like SAM.

**Questions:**

See weaknesses.

---

> ### Author Response · Authors · 2025-11-15
> **Our rebuttal to reviewer 2wTm**
>
> We sincerely thank you for your valuable time and constructive feedback on our paper and would like to address your concerns below.
>
> > **Q1: The efficiency of FusionFormer-L is no as good as other model settings.**
>
> **A1:** We completely agree that efficiency is a critical dimension. We would like to clarify FusionFormer-L's exceptional efficiency by discussing it from two perspectives: **theoretical complexity (FLOPs)** and **practical inference speed (FPS)**.
>
> In terms of theoretical complexity, FusionFormer-L demonstrates a clear leading advantage. As shown in Table 2, our model's FLOPs on Cityscapes are only 89.2 G, which is substantially lower than SCTNet-B (119.8 G) and SDPT-Small (131.3 G). This highlights the algorithmic efficiency of our MWFA design.
>
> Regarding practical inference speed (FPS), we acknowledge that SCTNet-B achieves a higher FPS. This is primarily due to a fundamental difference in architectural design: at inference time, SCTNet is a single-branch, purely convolutional network. This design allows it to fully leverage compute libraries that are highly optimized for standard convolutions, resulting in exceptional throughput on specific hardware. However, this speed-centric design also significantly constrains its representational power, especially on datasets with greater class diversity. For instance, on the ADE20K dataset, SCTNet-B only achieves 43.0% mIoU, which is substantially lower than the 46.7% achieved by our FusionFormer-L.
>
> A more compelling testament to our model's performance is the comparison with SDPT-Small, a SOTA attention-based model not specifically tailored for the real-time domain. On the ADE20K dataset, FusionFormer-L surpasses it in both accuracy (46.7% vs. 46.0%) and inference speed (110.6 FPS vs. 46.8 FPS).
>
> > **Q2: The proposed method is mainly demonstrated in segmentation task, the reviewer is wondering if the similar architecture can be extended to be used in foudation models like SAM.**
>
> **A2:** This is an excellent and forward-thinking question, and we sincerely appreciate you recognizing the potential impact of our work.
>
> To elaborate, creating a lightweight SAM-like model with our architecture would involve more than a simple backbone replacement. As demonstrated by pioneering works such as EfficientViT-SAM [1] and TinySAM [2], this task typically requires a systematic knowledge distillation framework. Within such a framework, one would retain SAM's same prompt encoder and mask decoder, while positioning our efficient FusionFormer as the student model's vision encoder to learn in a stage-wise manner from the original SAM teacher.
>
> While successfully applying FusionFormer in this paradigm would be a strong validation of its effectiveness, we also recognize that this direction is distinct from the core focus of our current paper. Nevertheless, we are grateful for your valuable suggestion, as it affirms the versatility and potential of our FusionFormer module. We are excited to explore this promising avenue in our future work.
>
> [1] Zhang Z, Cai H, Han S. Efficientvit-sam: Accelerated segment anything model without performance loss[C]//Proceedings of the IEEE/CVF Conference on Computer Vision and Pattern Recognition. 2024.
>
> [2] Shu H, et al. Tinysam: Pushing the envelope for efficient segment anything model[C]//Proceedings of the AAAI Conference on Artificial Intelligence. 2025.

---

### Official Review · Reviewer_NUdg · 2025-11-01

**Soundness:** 3
**Presentation:** 2
**Contribution:** 2
**Rating:** 2
**Confidence:** 4

**Summary:**

This paper introduces FusionFormer, a lightweight Transformer architecture tailored for real-time semantic segmentation. The primary contribution is the Multi-Window Fusion Attention (MWFA) module.

**Strengths:**

The overall architecture, including the backbone, light head, and VFM guidance, is clearly illustrated and described.

**Weaknesses:**

Let me briefly describe my guideline of reviewing the “efficient transformer” paper. Efficient transformers are hot about 3 or 4 years ago. Currently, without major novelty introduced or significant performance gain, it is hard to convince me that this paper is suitable for the whole community to read. As changing the backbone model will make re-training for everything, it does not make sense to use a new model without a major improvement.

- The paper's main claim to novelty is the MWFA module. However, it is difficult to see this as a major innovation. The core ideas—window-based attention , multi-scale processing , and adaptive token fusion —are all well-established and widely adopted concepts in the vision transformer community. The MWFA module appears to be an incremental combination of these existing techniques rather than a fundamental new approach to efficient attention.
- While the method achieves SOTA-level performance (claimed by the authors), the gains over previous work are not significant. These results are not a "significant performance gain" that would justify a new architecture in a crowded field, especially when the novelty is limited.
- In sufficient Ablation of the Core Architecture: The paper lacks a detailed ablation study on the MWFA module's design, making it hard to understand its properties or gain new insights. The specific multi-window configurations are presented without justification. A strong paper would ablate these choices, showing why this combination is optimal compared to others. This lack of deep architectural analysis makes the paper feel more like a report on a well-tuned system rather than a research paper providing new scientific insights.
- The paper's performance relied on VFM guidance, which is essentially a distillation technique. The ablation study on VFM guidance (Table 4 , Table 6 ) is more detailed than the ablation on the core architecture. On ADE20K, VFM guidance provides a major performance boost (e.g., 45.5 to 46.7 for FusionFormer-L, a 1.2 gain ). This makes it difficult to disentangle the contributions: is the performance from the novel MWFA architecture, or is it just showing that distilling from a powerful DINOv2 teacher is effective? The latter is quite obvious.


In conclusion, I think the contribution of this paper and the results are not significant. I know that doing large-scale experiments is hard. However, for such a crowd research area, it is necessary to make a contribution.

**Questions:**

NA

---

> ### Author Response · Authors · 2025-11-14
> **Our rebuttal to reviewer NUdg**
>
> We thank the reviewer for the detailed feedback. While we understand the reviewer's high standards for novelty and performance, we believe our paper offers valuable insights and a highly effective solution, and we address the reviewer's primary concerns below.
>
> # Main Comments
> > **Q1: On the Novelty of MWFA**
>
> **A1:** We thank the reviewer for pushing us to clarify our core novelty. While we build upon established concepts, we argue that our work represents a significant **conceptual shift** rather than a mere combination of existing techniques.
>
> Our primary contribution is not merely the MWFA module, but the entire framework that synergistically combines a novel architecture (FusionFormer) with VFM guidance. The innovation lies in our foundational design philosophy:
>
> - **Different design goal: Approximating global attention, Not just compressing it.** Many recent methods focus on aggressive computation compression (e.g., via extreme local or axial decomposition), **fundamentally diverging from the global attention of VFMs**. In contrast, our pursuit of "global approximation for better guidance" is a distinct and novel motivation.
>
> - **Purpose-driven architecture:** Every component within MWFA is designed to serve this unique goal. **Adaptive token fusion** within large windows reduces redundancy already captured by smaller windows, preserving efficiency; **head-aware enhancement** compensates for potential detail loss during fusion by leveraging the independence of attention heads; and **confidence-based weighting** across different windows allows adaptively focus on the most relevant spatial scales.
>
> - **A More General Guidance Framework.** Our approach offers a more general and elegant solution compared to methods that distill knowledge from larger, task-specific segmentation models (e.g.,Seaformer++ [1]).
>
> As self-supervised learning (like DINOv2) become the standard for building foundation models in domains with limited labeled data like medical imaging [2,3,4], our work provides a practical solution for resource-constrained applications.
>
> > **Q2: On the Significance of Performance Gains**
>
> **A2:** We respectfully disagree with the reviewer's assessment, as a closer examination of our results reveals a substantial advancement.
>
> On ADE20K, **FusionFormer-S** achieves 40.8% mIoU, surpassing all same-scale competitors like SeaFormer-S++ (39.7%) and SDPT-Tiny (39.4%), while operating at an impressive inference speed.
>
> **FusionFormer-B**'s performance is even more compelling. On Cityscapes, it reaches 80.2% mIoU, outperforming even larger models like SegNext-T (79.8%) and SeaFormer-L (79.4%). On ADE20K, with only 7.7M parameters, it achieves a remarkable 45.3% mIoU. This result surpasses all other models in the table except for the much larger SDPT-Small and our own FusionFormer-L.
>
> **FusionFormer-L** deliver maximum accuracy while remaining highly competitive in efficiency. On ADE20K, it sets a new SOTA with 46.7% mIoU, which surpasses other competitors like SCTNet-B (43.0%) by a large margin of +3.7%. While close to SDPT-Small (46.0%)—a model not designed for real-time scenario—FusionFormer-L is faster, running at ~2.4x the speed (110.6 FPS vs. 46.8 FPS).
>
> > **Q3: On the Ablation Study of MWFA: Justifying the Design Choices**
>
> **A3:** We appreciate the reviewer's call for a deeper architectural analysis and would like to clarify the detailed ablations and design principles behind MWFA.
>
> **Ablation Study on MWFA Components:** We respectfully draw the reviewer's attention to Table 3, which clearly demonstrates the contributions of the multi-window setting (MW), fusion weights ($\mathcal{F}$), head-aware fusion ($\mathcal{F}_r$), and window-level weighting ($\alpha_i$), validating our design.
>
> **Principled Justification for Multi-Window Configurations:** The configurations of $(w_i, r_i)$ were not chosen arbitrarily, but were guided by a core principle: the total complexity must be lower than standard window attention with w=8, while avoiding overly large $r$ that would diminish detail due to excessive fusion. For example, at a high-resolution stage using $w=\{ 8, 16, 32 \}$, the baseline of a standard window attention with $w=8$ is approximately $2Nd \cdot 8^2 = 2Nd \cdot 64$. From above, we chose corresponding fusion ratios of $ r = \{ 2, 2, 4 \}$. The resulting complexity is then calculated as: $2Nd \cdot (\frac{8^2}{2^4} + \frac{16^2}{2^4} + \frac{32^2}{4^4}) = 2Nd \cdot (4 + 16 + 4) = 2Nd \cdot 24$.
>
> [1] Wan Q, et al. SeaFormer++: Squeeze-enhanced axial transformer for mobile visual recognition[J]. International Journal of Computer Vision, 2025
>
> [2] Chen R J et al., Towards a general-purpose foundation model for computational pathology, Nature Medicine, 2024
>
> [3] Vorontsov E et al. A foundation model for clinical-grade computational pathology and rare cancers detection[J]. Nature medicine, 2024
>
> [4] Xu H, et al. A whole-slide foundation model for digital pathology from real-world data[J]. Nature, 2024

---

> > ### Author Response · Authors · 2025-11-14
> > **Our rebuttal to reviewer NUdg (continued)**
> >
> > Thank you for your detailed and thought-provoking review. Due to character limits, we are continuing our response in this comment. We will now address your final concern regarding the disentanglement of contributions from our architecture and the VFM guidance.
> >
> > > **Q4: Disentangling the Contributions of MWFA and VFM Guidance**
> >
> > The reviewer raises a crucial point: is the performance from MWFA or from VFM distillation? We argue that this is not an either/or question; rather, **the success of VFM guidance is direct proof of MWFA's effectiveness**. Our central thesis is that MWFA's design makes it uniquely adept at leveraging VFM guidance compared to prior efficient architectures.
> >
> > Our experiment in Table 4 (right) provides the definitive evidence. When subjecting different efficient attention mechanisms to the **exact same VFM guidance**, our MWFA-based model (46.7% mIoU) significantly outperforms architectures based on Window (45.2%), Axial (44.9%), and Conv (44.0%) attention. This experiment perfectly disentangles the two factors and proves the **inherent superiority of our MWFA architecture** in the context of VFM-guided segmentation.
> >
> > Furthermore, our models demonstrate strong standalone performance even without any VFM guidance.
> >
> > As shown in Table 6, on downstream segmentation tasks, FusionFormer-L achieves a highly competitive 45.5% mIoU on ADE20K, and FusionFormer-B achieves 80.0% mIoU on Cityscapes.
> >
> > Crucially, our **ImageNet-1K classification results (Table 1)**, which were obtained **without any VFM guidance**, also validate the superiority of our backbone. For example, FusionFormer-B achieves 79.3% Top-1 accuracy, outperforming strong baselines like SeaFormer-B (76.0%) and larger MiT-B1 (78.7%).
> >
> > In conclusion, we hope our detailed responses have addressed all your concerns and have better highlighted the novelty and significance of our work. We have taken your feedback very seriously. We are committed to making this paper as strong as possible and would be grateful for any specific guidance to strengthen our claims.
> >
> > Thank you once again for your valuable time and constructive feedback.

---

### Official Review · Reviewer_ggE5 · 2025-11-02

**Soundness:** 3
**Presentation:** 2
**Contribution:** 2
**Rating:** 4
**Confidence:** 4

**Summary:**

This paper introduces a new model for real-time semantic segmentation. This FusionFormer backbone is based on a lightweight adaptation of ViT foundation models. The method relies on one main architectural contribution: a learnable multi-scale token fusion scheme. This mechanism encodes information from deep layers of the foundation model by combining the aggregation of tokens from multiple window scales and a guidance distillation loss. Evaluations of the method on ADE20k and CityScape are presented as well as ablation studies.

**Strengths:**

The method shows consistent gains FPS and accuracy/mIoU on ImageNet-1k and two semantic segmentation datasets (ADE20k, Cityscape). The parameters overhead is low, and the performances are consistently above the presented baselines.

Each components of the method are well evaluated through dedicated ablations.

**Weaknesses:**

My main concern is related to the absence of comparison with foundation models dedicated to semantic segmentation, and in particular, SAM and SAM2. On natural images, these models tend to show similar performances to the ones presented here, with far better generalization capabilities.

The fusion strategy appears original here, but the idea to reduce the transformer's complexity with multi-scale fusion is not novel, with seminal work going back to SWIN and PVT backbones.

**Questions:**

What are the performances of the vanilla VFM + linear probing on ImageNet-1k?

It is not clear if the number of parameters reported also accounts for the parameters of the VFM.

---

> ### Author Response · Authors · 2025-11-13
> **Our rebuttal to reviewer ggE5**
>
> We thank the reviewer for the constructive comments. We provide our feedbacks as follows.
>
> # Main Comments
> >  **Q1 (Weakness 1): My main concern is related to the absence of comparison with foundation models dedicated to semantic segmentation, and in particular, SAM and SAM2.**
>
> **A1:** We sincerely thank the insightful feedback. It allows us to further elaborate on the core motivation and technical contribution of our work.
>
> Our primary motivation is to compensate for the performance degradation from the efficient designs of lightweight models in real-time segmentation. To this end, we propose FusionFormer, an architecture designed to approximate the global attention in VFMs, enabling more effective utilization of their powerful and general-purpose representations. The reasons for selecting a VFM like DINOv2 over SAMs for guidance are as follows:
>
> - **Avoiding Task-Specific Bias:** The vision encoder in SAM, while powerful, is initialized from a MAE-pretrained ViT and then further fine-tuned with a prompt-driven segmentation objective. In our targeted application scenarios, we aim to avoid the potential biases introduced by such task-specific training. Furthermore, self-supervised learning paradigms, exemplified by DINOv2, have become a mainstream approach for building general-purpose foundation models in domains with limited annotations, such as medical image analysis [1, 2]. In these fields, using SAMs would be inappropriate. Our proposed method, however, is directly applicable to these scenarios.
>
> - **Proven Generality and Superiority of Features:** As we noted in our introduction (lines 59-62), the remarkable performance of DINOv2 with a simple linear probe across various dense prediction tasks is a strong testament to the superiority and generality of its features, which makes it an ideal source of guidance.
>
> In summary, our work pioneers a solution for a distinct and important problem: enhancing real-time segmentation models via guidance from VFMs. The absence of a direct comparison with SAMs is thus a consequence of the fundamental disparity in their computational cost and application focus (real-time vs. zero-shot).
>
> > **Q2 (Weakness 2): The fusion strategy appears original here, but the idea to reduce the transformer's complexity with multi-scale fusion is not novel, with seminal work going back to SWIN and PVT backbones.**
>
> **A2:** While we agree that Swin and PVT were pioneers in multi-scale Transformers, our Multi-Window Fusion Attention (MWFA) introduces a novel mechanism for information fusion that is fundamentally different.The key distinction lies in how MWFA models multi-scale dependencies within a single layer, rather than across different stages. Specifically, our novelty is threefold:
>
> - **Multi-Window Processing:** MWFA processes multiple window sizes (e.g., 8x8, 16x16) in one layer, capturing diverse receptive fields simultaneously. This contrasts with Swin's fixed-size windows and PVT's stage-wise spatial reduction.
>
> - **Adaptive Token Fusion:** We introduce a learnable, content-aware fusion mechanism ($\mathcal{F}$ in Eq. 2) to dynamically aggregate tokens. This is a significant departure from the fixed, non-adaptive strategies, allowing reducing redundancy in larger windows.
>
> - **Head-Aware Enhancement:** Our fusion is further enhanced with a head-aware mechanism (Eq. 3), promoting spatial diversity across attention heads, which is a unique and fine-grained modulation.
>
> > **Q3:  What are the performances of the vanilla VFM + linear probing on ImageNet-1k?**
>
> **A3:** We have to clarify that the ImageNet-1K results in Table 1 are for our FusionFormer trained from scratch, **without VFM guidance**, to validate the effectiveness and efficiency of our proposed architecture.
>
> To directly answer the question, we provide the requested linear probing performance:
>
> | Model         | Parameters |   Top-1 Acc. % |
> |:--------------|:----------:|:---------------------------------------:|
> | DINOv2-S/14   | 21 M       | 81.1                                    |
> | DINOv2-B/14   | 86 M       | 84.5                                    |
> | DINOv2-L/14   | 300 M      | 86.3                                    |
> | DINOv2-g/14 | 1.1 B  | 86.5                                |
>
> > **Q4:  It is not clear if the number of parameters reported also accounts for the parameters of the VFM.**
>
> **A4:** We would like to clarify that the reported parameters, FLOPs, and FPS do not include the VFM.
>
> The VFM is exclusively part of a **training-only guidance branch**, which is entirely discarded during inference, explicitly labeled with "Training Only" in Figure 2. Consequently, it introduces no computational overhead to the final inference model.
>
> [1] Chen R J et al., Towards a general-purpose foundation model for computational pathology, Nature Medicine, 2024
>
> [2] Vorontsov E et al. A foundation model for clinical-grade computational pathology and rare cancers detection[J]. Nature medicine, 2024

---

### Author Response · Authors · 2025-11-29
**Summary of Contributions and Clarifications for the New Area Chair**

**To the Area Chair:**

We understand the significant workload caused by the recent administrative changes. To assist in your assessment, we provide this executive summary of our rebuttal. Our submission, **FusionFormer**, addresses a critical gap in real-time semantic segmentation by effectively bridging lightweight architectures with the powerful representations of Visual Foundation Models (VFMs).

We respectfully highlight the following key evidence from our rebuttal that directly addresses the reviewers' main concerns:

> **1. Decisive Rebuttal to "Incremental Innovation" Claims (Addressing Reviewer NUdg - Rating 2)**

**A1:** Reviewer NUdg raised a concern that our performance gains might stem solely from VFM distillation rather than architectural innovation. We firmly refute this with controlled experiments:

- **Architecture Matters (Table 4):** We compared FusionFormer against other efficient mechanisms (Window, Axial, Conv) under the exact same VFM guidance. FusionFormer consistently outperformed them (+1.5% to +2.7% mIoU). This proves that our Multi-Window Fusion Attention (MWFA) is structurally superior in utilizing VFM priors, and the success is not merely due to distillation.

- **Strong Standalone Performance:** Even without any VFM guidance, FusionFormer achieves competitive results on ImageNet-1K (Table 1) and segmentation benchmarks (Table 6), surpassing baselines like SeaFormer and MiT-B1. This validates the intrinsic value of our backbone design.

> **2. Clarification on Novelty vs. Existing Methods (Addressing Reviewer ggE5)**

**A2:** Regarding the comparison with Swin and PVT, we clarified that our novelty lies in **intra-layer multi-scale fusion**. Unlike Swin/PVT which process scales stage-wise, MWFA captures diverse receptive fields and dynamically aggregates tokens within a single layer. This design is specifically tailored to approximate global attention with linear complexity, which is crucial for real-time VFM adaptation.

> **3. Competitiveness against SOTA & Efficiency (Addressing Reviewers 2wTm & UZkc)**

**A3:**

- **Efficiency vs. Accuracy Trade-off:** We demonstrated that while some CNN-based models (e.g., SCTNet) may have higher FPS on specific hardware, they sacrifice significant accuracy (-3.7% mIoU compared to us). FusionFormer-L is the only model in this tier achieving >46% mIoU at >110 FPS on ADE20K.

- **Updated Baselines:** Regarding the suggested baselines (EdgeViT, FasterViT), we have added a discussion to clarify that they operate in a **significantly heavier computational regime** (e.g., FasterViT-0 is ~2.4x larger than our FusionFormer-L), and thus are **not direct competitors** in the lightweight real-time domain.

**Conclusion:** We believe the evidence provided in the paper and rebuttal demonstrates that FusionFormer is not just a combination of existing modules, but a robust, purpose-built solution for the emerging direction of VFM-guided efficient perception. For detailed, point-by-point responses to specific technical questions, please refer to the individual comments posted below each reviewer's thread. We hope this summary assists you in your final decision.

Sincerely, The Authors

---

### Meta-Review · Area_Chair_efu1 · 2025-12-23

**Summary:**

3 Reviewers (ggE5, NUdg and UZkc) all consider the novelty of the contribution is incremental. The proposed MWFA is supposed to work effectively with other measures like VFM guidance, which verifies the reviewers' worries. Besides, many  reviewers, especially the reviewer UZkc, asked for the comparison or discussion with the recent SOTA methods or the extension of the popular models like SAM,  which I also consider to be important while the corresponding rebuttal is weak.

**Reviewer Concerns:**

Just as the authors' message to AC indicates, the contribution or the novelty of the paper is the common concern among the reviewers. The novelty is incremental and thus hardly inspiring to the community though the improvements of experimental results are observable.

**Reviewer Scores:**

In terms of two highest score (6), one of the reviewers, i.e., Reivewer UZkc, still consider the paper to be limited in novelty after his reading the rebuttal while the other reviewer, i.e., Reviewer 2wTm, is a bit rash in giving a higher score after my reading the reviews. The lowest score is 2 from Reviewer NUdg who puts much emphasis on the novelty.

---

### Decision · Program_Chairs · 2026-01-26

Reject